

# Perceived physical activity during stay-at-home COVID-19 pandemic lockdown March–April 2020 in Polish adults

Stanisław H. Czyż[1,2,3] and Wojciech Starościak[1]

[1] Faculty of Physical Education and Sport, Wroclaw University of Health and Sport Sciences, Wrocław, Poland
[2] Faculty of Sport Studies, Masaryk University, Brno, Czech Republic
[3] Physical Activity, Sport and Recreation (PhASRec), North-West University, Potchefstroom, South Africa

Corresponding author
Stanisław H. Czyż,
stanislaw.czyz@awf.wroc.pl

## ABSTRACT

**Background:** Lockdowns amid the COVID-19 pandemic drastically reduced the possibility of undertaking physical activity (PA) in gyms, swimming pools, or work-related PA, *e.g.*, active commuting. However, the stay-at-home order could have reduced PA the most, *i.e.*, the ban of unnecessary outdoor activities. It affected free walking, running, skiing, active tourism, *etc*. It is, therefore, crucial to estimate how the stay-at-home order affected PA. We estimated how the stay-at-home order affected perceived PA and sedentary behavior compared to the pre-pandemic time in Poland.

**Methods:** We used a self-reported International Physical Activity Questionnaire— Long Form (IPAQ-LF) to estimate the time (minutes per day) of vigorous and moderate PA and walking and sitting time.

**Results:** We gathered data from 320 Polish participants. Bayesian approaches, including *t*-test and Bayesian correlations, were used to find differences and correlations between PA before and during the stay-at-home lockdown. Our data supported the hypotheses that vigorous PA, as well as walking, declined during the lockdown. Surprisingly, our data did not support the hypothesis that moderate physical activity was reduced. We found that moderate PA during lockdown increased compared to the pre-lockdown PA. As hypothesized, our data strongly evinced that sitting time inclined during the lockdown. PA decline was not correlated with the available living space. People who had access to gardens did not demonstrate a higher PA level than those without.

**Discussion:** Walking and sitting time have drastically changed during the stay-at-home lockdown, decreasing and increasing, respectively. Given results from studies focusing on lockdowns without the stay-at-home restriction, it may be assumed that letting people go outside is crucial in keeping them more active and less sedentary. Authorities should take into account the effect the stay-at-home order may have on PA and sedentary behavior and as a result, on health. Stay-at-home orders should be the last considered restriction, given its detrimental consequences.

## INTRODUCTION

COVID-19 pandemic lockdown policies have substantially reduced physical activity (PA) and increased sedentary behavior (*Stockwell et al., 2021*). International organizations and politicians argued that specific measures must be taken (*United Nations Department of Economic & Social Affairs, 2020*). As a result of these conjectures, PA recommendations during the COVID-19 pandemic have been widely published (*Hammami et al., 2020*; *Pinto et al., 2020*; *World Health Organization, 2020*). There are already a few estimations of how lockdowns affected the level of PA and how the inactivity caused by pandemic lockdowns may affect health (*King et al., 2020*).

Given that PA has been generally and globally decreasing (*Guthold et al., 2018*) and its insufficiency affects health in many dimensions (*Lee et al., 2012*; *Ekelund et al., 2016*), the question of whether lockdowns further reduced it is vital. A review by *Caputo & Reichert (2020)* included 41 studies related to PA and COVID-19 pandemic. They evinced that pandemic restrictions decreased PA. In another review, *Stockwell et al. (2021)* emphasized that most of the 66 reviewed studies demonstrated a decline in PA and an incline in sedentary behavior. These trends were observed regardless of the subpopulation studied or the methodology used (*Stockwell et al., 2021*). A survey on PA amid pandemic in Canada (*Rhodes et al., 2020*) showed that perceived moderate (*MPA*) to vigorous physical activity (*VPA*) decreased. *Meyer et al. (2020)* reported a significant decrease in PA in socially isolated participants (staying at home) in the USA. A cross-sectional study by *Qin et al. (2020)* showed that the number of Chinese with inadequate PA was doubled during the pandemic compared to the world prevalence before the pandemic.

Nevertheless, a few studies were showing the opposite, *e.g.*, participants who were able to go outside did not reduce their PA compared to the pre-pandemic activity (*Meyer et al., 2020*). Walking time (*Walking*) and *MPA* during the pandemic increased in Switzerland and France (*Cheval et al., 2020*). In Germany, sports-related activities declined while recreational increased (*Schmidt et al., 2020*).

Different approaches can partially explain these contradictory findings. Some studies were done with staying-at-home participants (stay-at-home order), some when outside activities, such as walking or running, were allowed. Moreover, some of the methods used to estimate PA during pandemic may be questioned. For instance, in a study using apps counting steps (*Tison et al., 2020*), PA was assessed based on smartphone applications. However, the probability of walking at home with a smartphone in hand seems rather dubious. Moreover, some people could have exercised at home without going out, *e.g.*, they could engage in strength training, yoga, or other activities not requiring much locomotion. Home-based training may efficiently reduce the negative impact of decreased PA due to pandemic restrictions, *e.g.*, a home-based resistance training proved to be a low-cost, efficient alternative to outdoor activities during the stay-at-home lockdown in Italy (*Vitale et al., 2020*).

Therefore, it is crucial to estimate PA and sedentary behavior relating to the specific context because of restrictions in a studied country. We aimed to assess PA during the most restrictive lockdown in Poland. Given all restrictions and results from the previous

studies, we advanced the hypotheses that *VPA*, *MPA*, and *Walking* will be much reduced during lockdown compared to the pre-lockdown period. However, the magnitude of that difference was unclear. We also advanced a hypothesis that sitting time (*Sitting*) before lockdown will be shorter than during lockdown.

## MATERIALS AND METHODS

### Study design

The study was approved by the Ethics Committee of the University School of Physical Education in Wrocław, Poland (no. 10.2020). All procedures were performed in accordance with the Declaration of Helsinki (*World Medical Association, 2013*).

All participants volunteered to participate in the study (by ticking the online box). Although the study was anonymous, *i.e.*, we did not collect any personal data, we asked participants to consent by clicking an appropriate checkbox at the beginning of the online survey. Given the nature of the study, we did not predetermine the sample size, *i.e.*, we tried to recruit as many participants as possible.

### Pandemic-related national health restrictions

The data was collected during the epidemic state between 31st March and 20th April 2020. The following restrictions affecting PA during the lockdown were in place:

- Free movement of people was restricted;
- People were allowed to leave home in precisely specified cases, such as for a trip to one's work or business, for essential trips to family and relatives for care or caregiving, shopping trips to purchase food and essential supplies, medicines or pet supplies, visits to the doctor or to deal with urgent official matters, volunteering related to coronavirus help, dog walk (only one person at a time);
- All public events were cancelled;
- All schools, universities, theaters, cinemas, cultural centers, restaurants, clubs, hairdressers, tattoo, beauty salons, gyms, sport clubs, libraries, zoo, public gardens, etc. were closed;
- Obligatory 14-day quarantine for all returning to the country;
- Foreigners were not allowed to enter the country;
- All flights from and to Poland, as well as other transport means (buses, trains, tourist border crossings), were canceled;
- Non-essential shopping centers and shops were closed;
- Children under 18 were not allowed to walk outside without a legal guide's assistance;
- Maximum five people during funerals and in churches were allowed;
- Access to parks, forests, boulevards, etc. was prohibited;
- Cars, bicycles, and other rent centers were closed;
- Limited personal contacts with the public or clients and between employees and reduction of employees in the workplace to the necessary minimum were allowed.

Penalties for not adhering to the restrictions policy were between 5 and 30 thousand PLN (about 1106–6640 EUR according to the National Polish Bank exchange course on 21st April 2020).

On 21st April, the restrictions were loosened, including the permission to walk outside for other than listed purposes, *e.g.*, for running, walking, cycling, etc. We decided to analyze the data gathered up to 21st April since IPAQ-LF is a 7-day recall questionnaire, which means the questionnaire completed on the 21st was referring to the lockdown period.

## Participants

Due to the total lockdown, the ways we could recruit participants were quite limited. We decided to attract potential participants *via* social media, including Twitter, Facebook (personal, university alumni, physical activity societies, media, and other profiles), private emails, and *via* radio (interviews in Meloradio and Radio Wrocław) and TV (interview in a regional public TVP 3). Interviews and a short description, and an invitation to participate in the study, were published on the media web pages.

We included all adults who were not quarantined due to the COVID-19 during the surveyed period (the last 7 days). Data of participants who did not tick the consent was not saved.

## Survey

The survey was anonymous and was done using Google Forms. The link: https://docs.google.com/forms/d/e/1FAIpQLSfeErFVyUGRav-05JeG_iE7hJ5ITVfceyjfxuQ7FlOE3RBf1Q/viewform?vc=0&c=0&w=1 was used for further dissemination. For convenience, we also used shorter links: https://tiny.pl/7qd7j and https://bit.ly/aktywnoscpolakow.

The survey consisted of three parts. In the first part of the survey, we asked to thick the informed consent box and collected demographic data about sex, age (date of birth), height, body mass, living place size (m$^2$), a number of household inhabitants, accessibility to own garden, and social status (schoolchild, student, white-collar worker, worker, pensioner, unemployed). Participants were able to add their comments too (indicating, *e.g.*, maternity leave).

In the second and third parts of the survey, we used IPAQ-LF. Since there is no more appropriate questionnaire than others (*Terwee et al., 2010*), we decided to apply IPAQ (*Craig et al., 2003*) as it is one of the most widely used questionnaires worldwide. It shows good reliability ($r = 0.81$) and is the most reliable (ICC = 0.97) physical activity questionnaire (*Helmerhorst et al., 2012*).

We chose the long-form (*Hagströmer, Oja & Sjöström, 2006*) as it assesses the PA level related to four domains: (1) during self-powered transportation, (2) at work, (3) during household and gardening work, and (4) during leisure time, including exercise and sport participation. *VPA* refers to activities that take hard physical effort and make you breathe much harder than normal. *MPA* refers to activities that take moderate physical effort and make you breathe somewhat harder than normal. It also allows collecting detailed information about sedentary activity.

We used the reliable Polish version of the questionnaire (*Biernat, Stupnicki & Gajewski, 2007*; *Biernat, 2013*). In the second part of the survey, we used IPAQ-LF to assess the PA level before the pandemic. We asked about seven typical days in January 2020. Although the IPAQ was not ensured to be so much retrospective and the reliability and validity of 3–4 months recall questionnaire could be doubted, we decided to collect the data nevertheless. The third part of the survey consisted of IPAQ-LF dedicated to assess PA level during lockdown (7 days recall).

## Data processing and truncation

The time of PA was calculated according to the "*Guidelines for data processing analysis of the International Physical Activity Questionnaire (IPAQ)—Short and long forms*" (*Sjöström et al., 2005*). We processed the data following the rules of data cleaning, unreasonably data exclusion, minimum values inclusion, and truncation of data described by *Sjöström et al. (2005)*.

Based on the assumption that an average of 8 h per day is spent sleeping (*Sjöström et al., 2005*), five participants were excluded from further analysis. Their sums of all *Walking*, *MPA*, and *VPA* time variables were greater than 960 min (16 h) (*Sjöström et al., 2005*).

A total of 152 records of 111 participants were truncated, *i.e.*, the variables total *Walking*, total *MPA*, and total *VPA* were calculated, and then, if any of them exceeded 180 min (3 h), they were truncated to 180 min.

Given the *VPA*, *MPA*, and *Walking* are multiplied by different coefficients in each PA domain, *i.e.*, occupational, transport, household, and leisure-related PA, it was impossible to use the truncated time to calculate metabolic equivalents-minutes (MET). Instead of using METs as initially proposed by *Sjöström et al. (2005)* we report the time (minutes per day) for each activity (*VPA*, *MPA*, and *Walking*). Time spent in *VPA*, *MPA*, and *Walking* are widely used measures of PA, *e.g.*, they are used by *World Health Organization (2010)*.

## Data analysis

We report descriptive statistics on *VPA*, *MPA*, *Walking*, and *Sitting* time (min/day).

Bayesian one-sided *t*-test was used to compare PA before and during the lockdown. We assumed that *VPA*, *MPA*, *Walking* before lockdown > *VPA*, *MPA*, *Walking* during lockdown. The opposite hypothesis was advanced concerning the sitting time, *i.e.*, *Sitting* before lockdown < *Sitting during* the lockdown.

The null hypothesis (H$_0$) assumed an absence of differences between the pre-lockdown and lockdown measurements, whereas the alternative hypothesis (H$_1$) assumed a directional difference.

Bayes Factor (BF) was used to provide the alternative hypothesis's quantified evidence relative to the null hypothesis. The evidence categories for Bayes factor (BF) were set as proposed by *Jeffreys (1998)*, *Wetzels et al. (2011)*.

A Cauchy scale parameter of 0.707 (prior) was assigned for the effect size (*Rouder et al., 2009*).

**Table 1 Descriptive statistics.**

| | Age (years) | | Height (cm) | | Body mass (kg) | | Living place size ($m^2$) | |
| --- | --- | --- | --- | --- | --- | --- | --- | --- |
| | Female | Male | Female | Male | Female | Male | Female | Male |
| Valid | 197 | 121 | 197 | 123 | 197 | 123 | 197 | 121 |
| Mean | 32.858 | 35.95 | 166.198 | 180.374 | 62.423 | 81.39 | 95.308 | 91.727 |
| SD | 12.001 | 13.83 | 5.848 | 5.805 | 10.047 | 11.867 | 61.645 | 67.937 |
| Minimum | 18 | 17 | 150 | 167 | 42 | 59 | 24 | 10 |
| Maximum | 76 | 75 | 180 | 194 | 98 | 130 | 400 | 413 |

Bayes factor robustness checks were performed. We used JASP specified wide and ultra-wide priors to examine the extent to which our conclusions depend on our prior specification (*van Doorn et al., 2020*). They display the Bayes factor as a function of the width of the Cauchy prior on effect size using the wide (scale = 1) and ultrawide prior (scale = $\sqrt{2}$).

Additionally, we performed pairwise Bayesian Pearson correlation tests.

The Bayesian statistics were performed using JASP (version 0.14.1).

## RESULTS

### Study population

There were 326 records registered in Google Forms between 8–21 April 2020. Data from one participant was removed as she was 14 years old, whereas the IPAQ-LF is dedicated to adults. Records of 320 participants were analyzed. Three participants were quarantined. Out of 320 participants, 10 participants (six females, four males) indicated they had contraindications (as advised by a medical doctor) to exercise. 45.62% of the participants (*n* = 146, 93 women, 53 men) declared they had gardens for their own use.

Some of the records were incomplete; however, it did not disqualify them from further analysis, *i.e.*, the missing data referred to the body height or size of the living place.

Descriptive statistics are presented in Table 1. Frequencies of the participants' social status are presented in Table 2.

### Physical activity before and during the lockdown

Results of the Bayesian *t*-test for *VPA*, *MPA*, and *Walking* are presented in Table 3.

Our hypothesis that *VPA* time before the lockdown was longer than during lockdown was anecdotally (BF = 1.48) supported by our data (Fig. 1). Although the time spent on *VPA* before the lockdown was longer, the difference was slightly above 9 min/day. Bayes factor robustness check with wide prior also anecdotally supported $H_1$ (BF = 1.065). On the other hand, our data anecdotally supported $H_0$ while the prior distribution was set to ultrawide (BF = 0.76).

In contrast, our data negatively supported the hypothesis that *MPA* before the lockdown was longer than during the lockdown (see Table 3 and Fig. 2). The evidence against the alternative hypothesis $H_1$ was strong (BF = 0.046). It is not surprising since the average time spent on *MPA* was longer during than before lockdown by about 2 min/day (Table 4).

**Table 2 Frequencies (*n*) of participants' social status.**

| Sex | Social status | Frequency | Percent | Cumulative percent |
|---|---|---|---|---|
| Female | Farmer | 1 | 0.508 | 0.508 |
| | Maternity leave | 4 | 2.03 | 2.538 |
| | Pensioner | 7 | 3.553 | 6.091 |
| | Schoolchild | 7 | 3.553 | 9.645 |
| | Student | 63 | 31.98 | 41.624 |
| | Unemployed | 6 | 3.046 | 44.67 |
| | White-collar worker | 93 | 47.208 | 91.878 |
| | Worker | 16 | 8.122 | 100 |
| | Total | 197 | 100 | |
| Male | Farmer | 0 | 0 | 0 |
| | Maternity leave | 0 | 0 | 0 |
| | Pensioner | 3 | 2.439 | 2.439 |
| | Schoolchild | 6 | 4.878 | 7.317 |
| | Student | 28 | 22.764 | 30.081 |
| | Unemployed | 1 | 0.813 | 30.894 |
| | White-collar worker | 75 | 60.976 | 91.87 |
| | Worker | 10 | 8.13 | 100 |
| | Total | 123 | 100 | |

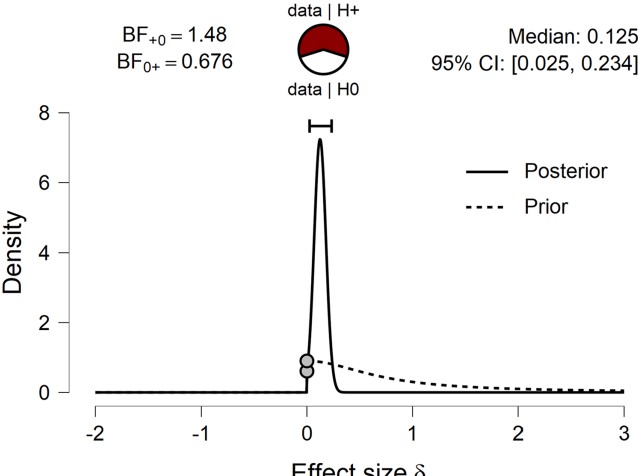

Legend:
CI – confidential intervals.
Pizza plot: Data|H0 – data supporting null hypothesis; Data|H+ - data supporting alternative hypothesis.
$BF_{+0}$: Bayes factor that quantifies evidence for the one-sided alternative hypothesis that group one > group two, relative to the null hypothesis.; $BF_{0+}$: Bayes factor that quantifies evidence for the null hypothesis, relative to the one-sided alternative hypothesis that group one > group two.

**Figure 1 Prior and posterior *VPA*.** Displays the prior (dashed line) and posterior (solid line) distribution of the effect size under the alternative hypothesis; the gray circles represent the height of the prior and the posterior density at effect size delta = 0. The horizontal solid line represents the width of the 95% credible interval of the posterior.

**Table 3 Bayesian paired sample *t*-test results for *VPA*, *MPA*, and *Walking* before and during the lockdown.**

| Activity | Comparison | N | Mean (min/day) | SD | SE | 95% Credible interval | | Bayesian paired samples *t*-test | |
|---|---|---|---|---|---|---|---|---|---|
| | | | | | | Lower | Upper | $BF_{+0}$ | Error % |
| **VPA** | Before lockdown | 320 | 56.725 | 53.1 | 2.968 | 50.885 | 62.565 | 1.48 | ~7.046e−8 |
| | During lockdown | 320 | 47.362 | 50.88 | 2.844 | 41.767 | 52.958 | | |
| **MPA** | Before lockdown | 320 | 66.394 | 58.328 | 3.261 | 59.979 | 72.809 | 0.046 | ~0.001 |
| | During lockdown | 320 | 68.375 | 59.197 | 3.309 | 61.864 | 74.886 | | |
| **Walking** | Before lockdown | 320 | 72.431 | 62.22 | 3.478 | 65.588 | 79.274 | 26.804 | ~2.265e−4 |
| | During lockdown | 320 | 56.528 | 59.154 | 3.307 | 50.022 | 63.034 | | |

Note:
The alternative hypotheses specify that VPA, MPA, and Walking time before lockdown are longer than during lockdown for all tests. SD, standard deviation; SE, standard error of the mean. $BF_{+0}$: Bayes factor that quantifies evidence for the one-sided alternative hypothesis that group one > group two, relative to the null hypothesis.

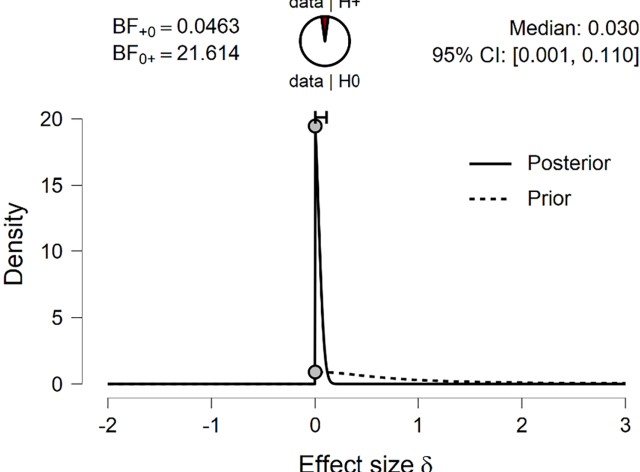

Legend:
CI – confidential intervals.
Pizza plot: Data|H0 – data supporting null hypothesis; Data|H+ - data supporting alternative hypothesis.
$BF_{+0}$: Bayes factor that quantifies evidence for the one-sided alternative hypothesis that group one > group two, relative to the null hypothesis.; $BF_{0+}$: Bayes factor that quantifies evidence for the null hypothesis, relative to the one-sided alternative hypothesis that group one > group two.

**Figure 2 Prior and posterior *MPA*.** Displays the prior (dashed line) and posterior (solid line) distribution of the effect size under the alternative hypothesis; the gray circles represent the height of the prior and the posterior density at effect size delta = 0. The horizontal solid line represents the width of the 95% credible interval of the posterior.

**Table 4 Bayesian paired sample *t*-test results for *Sitting* time before and during the lockdown.**

| Activity | Comparison | N | Mean (min/day) | SD | SE | 95% Credible interval | | Bayesian independent samples *t*-test | |
|---|---|---|---|---|---|---|---|---|---|
| | | | | | | Lower | Upper | $BF_{-0}$ | Error % |
| **Sitting** | Before lockdown | 320 | 276.306 | 168.538 | 9.422 | 257.77 | 294.843 | 866.38 | ~ 8.902e−9 |
| | During lockdown | 320 | 386.087 | 420.157 | 23.487 | 339.878 | 432.297 | | |

Note:
The alternative hypothesis specifies *Sitting* time before the lockdown was shorter than during lockdown. SD, standard deviation; SE, standard error of the mean. $BF_{-0}$: Bayes factor that quantifies evidence for the one-sided alternative hypothesis that group one < group two, relative to the null hypothesis.

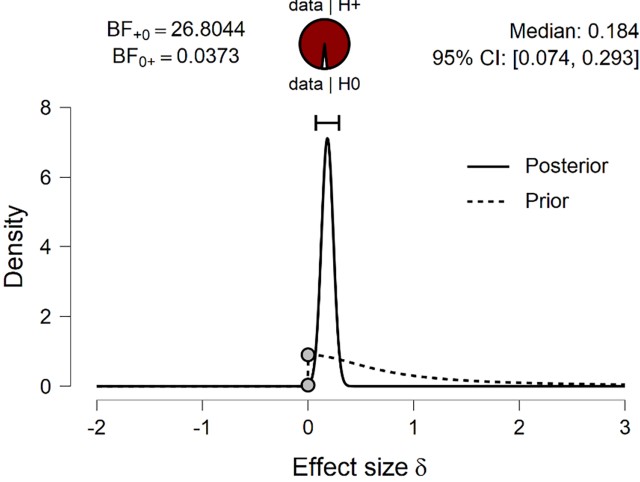

Legend:
CI – confidential intervals.
Pizza plot: Data|H0 – data supporting null hypothesis; Data|H+ - data supporting alternative hypothesis.
BF+0: Bayes factor that quantifies evidence for the one-sided alternative hypothesis that group one > group two,
relative to the null hypothesis.; BF0+: Bayes factor that quantifies evidence for the null hypothesis, relative to the
one-sided alternative hypothesis that group one > group two.

**Figure 3 Prior and posterior *Walking*.** Displays the prior (dashed line) and posterior (solid line) distribution of the effect size under the alternative hypothesis; the gray circles represent the height of the prior and the posterior density at effect size delta = 0. The horizontal solid line represents the width of the 95% credible interval of the posterior.

The Bayes factor robustness check with wide and ultrawide priors also supported $H_0$ (BF = 0.032; BF = 0.023; respectively).

Our data strongly supported $H_1$ related to *Walking* time (BF = 26.804). We assumed that *Walking* time before the lockdown was shorter than during lockdown. As shown in Table 3 and Fig. 3, the mean *Walking* time decreased from 72.431 min before lockdown to 56.528 min/day during the lockdown. Our data still strongly supported $H_1$ when the prior distribution was set to wide and ultrawide (BF = 19.62; BF = 14.12; respectively).

## Sedentary behavior

We tested our $H_1$ related to sitting time, assuming that *Sitting* time before lockdown will be shorter than during lockdown. The $H_1$ was strongly (decisively) supported by our data (Fig. 4). The BF = 866.38 (see Table 4). The S*itting* time during lockdown increased by an average of 40%, *i.e.*, by almost 100 min/day compared to the pre-lockdown period. Our data extremely strongly supported $H_1$ also with wide (BF = 646.5) and ultrawide prior (BF = 470.4) (Fig. 5).

## *VPA, MPA, Walking*, and *Sitting* time correlates

One of the most bothersome restrictions during the lockdown was to stay-at-home order. People could have been physically active only in their houses and gardens unless still working professionally. Therefore, we decided to see if there were any correlations between the size of their living places ($m^2$) and the number of inhabitants sharing a living place. We divided the living place size by the number of inhabitants, and we correlated it with *VPA, MPA, Walking*, and *Sitting* time. We performed pairwise Bayesian Pearson
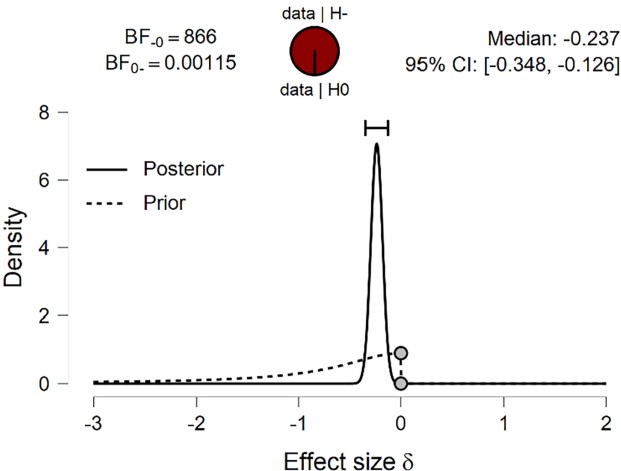

**Figure 4 Prior and posterior of *Sitting*.** Displays the prior (dashed line) and posterior (solid line) distribution of the effect size under the alternative hypothesis; the gray circles represent the height of the prior and the posterior density at effect size delta = 0. The horizontal solid line represents the width of the 95% credible interval of the posterior.

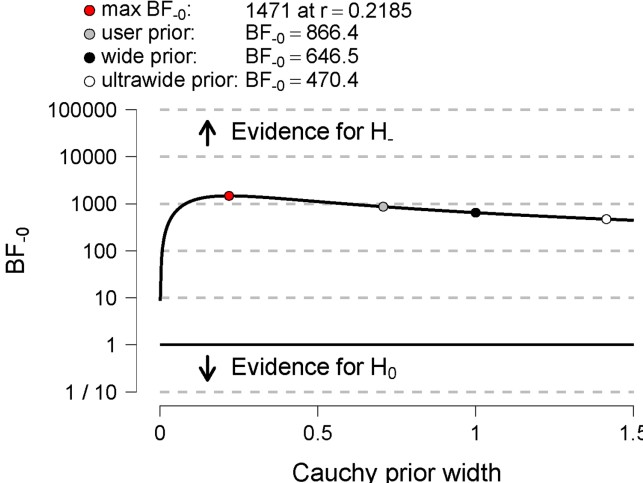

**Figure 5 Bayes factor robustness check for *Sitting*.** Displays the Bayes factor as a function of the width of the Cauchy prior on effect size. The black circle represents the Bayes factor computed with a wide prior; the white circle represents the Bayes factor computed with an ultrawide prior; the gray circle represents the Bayes factor computed with the user-defined prior.

correlation tests. We tested $H_1$ that the available living space ($m^2$/person) is positively correlated with *VPA*, *MPA*, *Walking* against $H_0$, assuming there is no such correlation. The $H_1$ was anecdotally supported by our data ($r = -0.052$, BF = 0.107; $r = -0.007$, BF = 0.071; $r = -0.016$, BF = 0.073; for *VPA*, *MPA*, *Walking* accordingly).

**Table 5** Bayesian independent sample *t*-test results for *VPA, MPA, Walking* time during the lockdown in participants with and without own gardens.

| Activity | Comparison | N | Mean (min/day) | SD | SE | 95% Credible interval | | Bayesian independent samples *t*-test | |
|---|---|---|---|---|---|---|---|---|---|
| | | | | | | Lower | Upper | $BF_{-0}$ | Error % |
| VPA during lockdown | No garden | 173 | 44.434 | 47.805 | 3.635 | 37.259 | 51.608 | 0.321 | ~ 1.165e−4 |
| | Own garden | 146 | 49.925 | 53.35 | 4.415 | 41.198 | 58.651 | | |
| MPA during lockdown | No garden | 173 | 64.277 | 58.072 | 4.415 | 55.563 | 72.992 | 0.456 | ~ 1.145e−5 |
| | Own garden | 146 | 72.466 | 59.885 | 4.956 | 62.67 | 82.261 | | |
| Walking during lockdown | No garden | 173 | 58.624 | 60.726 | 4.617 | 49.511 | 67.737 | 0.072 | ~ 5.130e−6 |
| | Own garden | 146 | 53.199 | 56.59 | 4.683 | 43.942 | 62.455 | | |

Note:
For all tests, the alternative hypothesis H1 specifies that the group "no garden" is less than group "own garden". SD, standard deviation; SE, standard error of the mean. $BF_{-0}$: Bayes factor that quantifies evidence for the one-sided alternative hypothesis that group one < group two, relative to the null hypothesis.

Regarding the *Sitting* time, we advanced the opposite hypothesis ($H_1$) that that available living space ($m^2$/person) is negatively correlated with *Sitting* time. Likewise, this hypothesis was anecdotally supported by our data ($r = -0.077$, BF = 0.031).

To assess whether participants with their garden were more active than those without, we applied the Bayesian *t*-test for *VPA*, *MPA*, *Walking*, and *Sitting* time. We tested $H_1$ that garden owners will be more physically active than the participants without gardens. All three tests did not yield supportive evidence toward these hypotheses (Table 5).

On the other hand, we hypothesized that participants without their garden would be sitting longer than garden owners. Conversely, our data strongly supported $H_0$, *i.e.*, the groups were not different in this regard (BF = 0.077). Garden owners spent on average 405.589 ± 552.521 min/day sitting during lockdown while those without gardens 371.803 ± 262.102.

## DISCUSSION

Our study aimed to assess the physical activity level during the total pandemic lockdown in Poland. We advanced hypotheses that *VPA*, *MPA*, and *Walking* time will be much reduced during lockdown compared to the pre-lockdown period and that sitting time (*Sitting*) before lockdown will be shorter than during lockdown.

As hypothesized, we found that *VPA* and *Walking* time decreased during lockdown compared to the pre-lockdown period. On the other hand, *MPA* increased during the lockdown. Our data extremely strongly supported the hypothesis that *Sitting* time during lockdown increased.

Our results align with general trends reported by other authors (*Caputo & Reichert, 2020*; *Stockwell et al., 2021*) though increased *MPA* was a rather unexpected finding and against the advanced hypothesis. This finding has to be considered with caution. As *Biernat & Piątkowska (2016)* noticed, there is a tendency to overestimate PA in general. *MPA* is responsible for 87.5% of overestimations. Moreover, women tend to overestimate MPA in the household domain significantly. Given our sample consisted of 197 females and 123 males, this tendency to overestimate *MPA* in women could be enhanced by the unequal female/male ratio. Another explanation may relate to the

characteristic of *MPA*. Since DIY (do-it-yourself) stores were open during the stay-at-home lockdown, many people started to renovate apartments and houses. MPA could have been, therefore, associated with the at-home work. However, this speculation has not been supported yet.

We also found that during the stay-at-home lockdown, most participants satiated WHO recommendations (*World Health Organization, 2010*), *i.e.*, they were engaged in more than 150 min of *MPA* per week or 75 min of *VPA* per week (or a combination of these two). Only 32 participants (10%) did not meet the recommended PA time. On the other hand, PA during lockdown was reduced compared to the pre-pandemic period. Our findings are consistent with most previous studies on PA during the pandemic (*Stockwell et al., 2021*). A total of 29 participants did not meet the WHO recommendation before the pandemic. It is 9.06% of the sample size. Both numbers, *i.e.*, 10% during the lockdown and 9.06% before lockdown, of physically inactive people have to be considered cautiously. Although there are no recent studies on inactivity prevalence in Poland, the available data suggest that around 30–39.9% of Polish have insufficient PA (*Guthold et al., 2018*).

The fact that most of the participants met the WHO recommendations is vital as PA may decrease the severity of symptoms and risk of mortality due to COVID-19 (*Hudson & Sprow, 2020*). However, PA is not the only factor worth taking into account; another is *Sitting*. In our opinion, authorities and decision-makers should consider the stay-at-home order as the last, the most detrimental restriction. As measured by the *Sitting* time, sedentary behavior has increased by almost 100 min/day, reaching 386 min/day on average. It means that our participants spent more than 6 h sitting per day. Our data extremely strongly supported the hypothesis about an incline in sedentary behavior, and this finding is consistent with previous studies (*Stockwell et al., 2021*).

The limitations of our study have to be acknowledged. The sample is not representative. Given, we advertised our study among our friends and through social media devoted to physical activity, sports, etc., our sample may be more representative of the physically active part of the Polish population than the whole population. Lastly, we used a subjective tool, *i.e.*, a self-reported questionnaire. It is widely reported that the PA level estimated based on self-reported questionnaires is usually overestimated compared to the more objective measures (*Biernat & Piątkowska, 2016*).

This paper's strength is that it is one of the very few studies on PA during the stay-at-home lockdown. Although lockdowns affected many countries' economies around the world, the restrictions introduced by many governments differed. Canceling flights, closing hotels, gyms, or swimming pools undoubtedly could have reduced PA. However, people who were not obliged to stay at home could have taken up other outdoor activities such as cycling, skiing, running, trekking, or any other. It is the stay-at-home order which probably made the PA less accessible in general.

It is worth emphasizing that we estimated PA at the very beginning of the pandemic, *i.e.*, the data was gathered between 8 and 21 April 2020. People are used to pandemic regulations, and stay-at-home restrictions are not so easily ordered anymore, so it is pretty

plausible that PA is not so much reduced anymore. This issue could be addressed in subsequent studies.

## CONCLUSIONS

In summary, *VPA* and *Walking* during stay-at-home lockdown declined. *MPA* and *Siting* increased. PA decline was not correlated with the available living place. People who have access to gardens did not demonstrate a higher PA level than those without.

We strongly recommend re-consider the negative effects the stay-at-home lockdowns have on *VPA* and *Walking*, and as an effect, on health. Perhaps, allowing citizens to walk outside and let them exercise outdoor would increase both: *Walking* and *VPA*, and at the same time, it would decrease time spent sitting. As a result, a detrimental effect the stay-at-home order has on health could be mitigated. Decision-makers should also consider the harmful effects of extended sitting since it increases the risk of death from cardiovascular disease and cancer.

## ACKNOWLEDGEMENTS

We would like to thank all disseminating details and invitations to our study, specifically, our Master students and local media. The results were partially used in the students' Master dissertations (all of them in Polish). Frequentists statistics were used to analyze the data.

### Funding

The authors received no funding for this work.

### Competing Interests

The authors declare that they have no competing interests.

### Author Contributions

- Stanisław H. Czyż conceived and designed the experiments, performed the experiments, analyzed the data, prepared figures and/or tables, authored or reviewed drafts of the paper, and approved the final draft.
- Wojciech Starościak performed the experiments, authored or reviewed drafts of the paper, and approved the final draft.

### Human Ethics

The following information was supplied relating to ethical approvals (*i.e.*, approving body and any reference numbers):

Ethics Committee of the Wroclaw University of Health and Sport Sciences, Poland granted approval to carry out the study (no. 10.2020).

## Data Availability

The data is available at OSF: https://osf.io/syrhn.

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
