# Peer review of "Perceived physical activity during stay-at-home COVID-19 pandemic lockdown March–April 2020 in Polish adults"

_PeerJ, doi:10.7717/peerj.12779_

## Round 0.1 · original submission · Major Revisions

Please carefully review the paper expetially in introduction and conclusion sessions. The revisors highlighted several modifications to consider,


[Reviewer 1 ·

Basic reporting

INTRODUCTION
Please, lengthen and deepen the introduction. some studies that could help you are:
1. https://pubmed.ncbi.nlm.nih.gov/21291246/ to comment the difference between outdoor activity and indoor (lockdown) activity
2. https://pubmed.ncbi.nlm.nih.gov/31077254/ physical activity and motivation
3. https://pubmed.ncbi.nlm.nih.gov/34302530/ Child factors associated with mental health and well-being during COVID-19
4. https://pubmed.ncbi.nlm.nih.gov/33588981/ Impact of COVID-19 Lockdown on physical activity and lockdown
5. https://pubmed.ncbi.nlm.nih.gov/33311526/


Line 56-57, 60. Is this writing reference correct, or is it more accurate to insert only a year? E.g. Jafari et al. (2021) show that etc.
Please check the editorial rule

Line 63-64. Please insert which study

Line 65. Walking between brackets, it is not an abbreviation

Line 71-76. Please, think to rewrite the goal of the study in a more explicit manner, avoiding inserting part of material and methods (e.g. IPAQ)

MATERIAL AND METHODS
Line 81-82. Please explain and insert references

Line 84-85. Please could you explain? When it was writing the paper, before or after that, you have collected data? This concept was not clear

Line 114-116. I think that it could be more ordinated if you insert a section to describe and explain the IPAQ questionnaire

DATA ANALYSES
Please, consider writing this section (Data analyses) again in a more precise manner

DISCUSSION
Discussion is a reply to results. Please discuss more deeply the results using literature

Experimental design

no comment

Validity of the findings

no comment

Reviewer 2 ·

Basic reporting

The introduction section needs the insert of a more supportive rationale and required a better explanation of many arguments. I suggest including more information about the health negative effects of augmented sedentary behavior and the positive benefits of an active lifestyle. I suggest adding an explanation about the different exercise intensities, how is important to measure vigorous and moderate PA, and the walking and sitting time. Then, I suggest paying attention to the lockdown period and the effects it has on the population, especially young adults, and adults. Several studies investigated these peculiar aspects.
The hypothesis of the study is not well clear, and the aim of the study is not supported by a logical and clear background. This section needs a complete rewrite.
Line 73-76: I suggest moving this information from the introduction section to the method section.

All the figures needed a clear legend of the acronyms at the bottom. Moreover, they need the insert of significance values or not. The captions are shown. Tables need partial corrections. I suggest deleting the “missing data”, “minimum” and “maximum” from the table and linking mean+SD.

Experimental design

The scope of this paper fit with the aims and scope of the journal.
Method section required some corrections.
I suggest moving the lockdown restrictions in a supplementary file to lighten the reading and maintain only the most important restriction related to the PA level. I suggest removing the link of the survey from lines 131 to 135 and maintaining only the shorter link. I suggest checking the unit of measure through the paragraphs. I think it would better explain why the minutes of these activities were truncated to 180 minutes if the activity was higher than 180 minutes (line 170). The inclusion and exclusion criteria are not well reported. I suggest adding them.
The sample is not representative of the entire Poland population, so the results cannot be generalized. I suggest refining the sample such as active people or a specific area of Poland.

Validity of the findings

I suggest rewriting this section explaining the results of the study and not only reporting them. An accurate comparison with the findings of other researchers is needed. More explanation and justification of your results are required.
In the end, I suggest adding some tips to reduce sedentary behavior during the lockdown period or to return to an active lifestyle in a safe way.

Additional comments

I would thank the author for this paper, the aim of this study was sure interesting. The unique period that we all are living needs evaluation and consideration. Before publication, this work needs many corrections.
In general, this manuscript showed several mistakes that required revisions. The introduction section and the discussion section showed a lack of clarity and logical paragraphs constructs. English languages need a relevant revision. Many phrases are not supported by recent literature. Then, I suggest to re-write the abstract in accordance with the corrections.

Reviewer 3 ·

Basic reporting

investigating the perceived physical activity during the stay-at-home lockdown March-April 2020 in Poland. I really appreciate the effort of the authors working in this field, because there are few evidences regarding the consequences of lockdown and social distances on physical activity and its impact on health. From a methodological standpoint, the research was well conducted. However, several issues need to be fixed to improve the general quality of the manuscript. At this stage, my indication is to request a support of a Native English speaker to improve the general readability of the article.

Lines from 46 to 76. In my opinion, the introduction need to be more hypothesis driven to allow readers to see the basis of your study. It is important to be clear what the practical question was trying to address to help readers to understand your study. The key issue here is to set-up the experimental approach to your study.

Line 53. I would suggest to insert here as a paragraph the part relating to the Poland National Health restrictions (Lines from 87 to 116), in order to give to the readers what was the situation in Poland.

Lines from 67 to 69. I would suggest to include here as a reference a recent article of Vitale et al, 2020 (PMID: 33352676) who evaluated the effect of a six-month home-based resistance-training program on muscle health and physical performance in healthy older subjects during the unique condition of home confinement caused by the COVID-19 pandemic in Italy.

Lines from 71 to 76. My suggestion is to define pre-specified primary and secondary aims. This will help readers to understand the research question that you are trying to address.

Experimental design

The methods needs to be improved so the study can be replicated as to equipment, subjects, context of physical activity level and rationale for the design of each dependent and independent variable. Therefore, readers needs to know more about enrolment, subjects procedures etc.

Lines from 80 to 85. I would suggest to call this paragraph “Study design”. Please, describe also in this paragraph the study design and the experimental approach to the problem. In accordance to the pre-specified primary and secondary aims of your study, you have to describe here the primary and secondary outcome measures, including how and when they were assessed. At this point a flowchart of the study will help readers to understand the temporal order of your study.

Lines 81 and 82. I would suggest adding here “all procedures were performed in compliance with laws and regulations governing the use of human subjects (Declaration of Helsinki)”. Please check it over.

Line from 82 to 84. I would suggest adding here “All subjects received explanation of purpose, methods, potential risks and benefits of the study and written informed consent was obtained from all participants”.

Lines from 118 to 124. Please place here all the information regarding the inclusion and the exclusion criteria of subjects’ participation in your study.

Line 125. I would suggest to move “There were 326 records registered in Google Forms between 08 – 21 April 2020” in the results section.

Line 126. I would suggest to move “The survey was anonymous.” In the survey section.

Line 125 and 126. I would suggest to move “All participants 126 volunteered to participate in the study (by ticking the online box).” In the study design section.

Line 127 and 128. I would suggest to move “Data from one participant was removed as she was 14 years old, whereas the IPAQ-LF is dedicated to adults.” In the result section.

Validity of the findings

In your result section, please insert only the raw results and the specific p values without discussions or speculations.
I would suggest creating only one table about the participants’ characteristics at baseline. Specifically, please merge Table 1 and Table 2 together.

I would suggest creating only one table regarding the Bayesian t-test results. Specifically, please merge Table 3, Table 4 and Table 5.

I would suggest creating a panel with your Figure 1, Figure 2, Figure 3 and Figure 4.

Line 208. I would suggest calling this paragraph “Study population”.

Line 210. Could you please specify here which the contraindication to exercises were?

Line 213. Could you please justify why you did not disqualify the missing data from your analysis?

Lines from 293 to 338. The discussion needs to reflect what you found, how it relates to the literature and then what it means physiologically or from a practical aspect and each paragraph should be in a logical sequence as at present it is a bit hard to follow. The clarity of your discussion needs to be improved a bit and qualified where appropriate as you need to stick to what your experimental design can tell us with your data and limit speculation or qualify them, and make sure that your statement are referenced.

Lines 319 and 320. I would suggest to delate the first limitation of your study. In fact, I understand that you have performed an observational study which does not need a randomisation process.

Line 341. I would suggest inserting in your conclusions the practical application of your study. What should now physicians, trainers and practitioners now have to do after reading your paper? Does it affect practice is the key factor for this section.

Additional comments

Line 55. I think there is a sort of typo at the beginning of the line.

---

## Round 0.2 · accepted · Accept

Please take note of the comment of reviewer 2 and correct the conclusion section.

Reviewer 1 ·

Basic reporting

I think it's appropriate and clear
I would not make any recommendations to that effect.

Experimental design

the research question now is well defined, and the research is original.
Further, aims are clear

Validity of the findings

Paper talk about the pandemic period, it is interesting and now the conclusion supports the results

Reviewer 2 ·

Basic reporting

No comment

Experimental design

No comment

Validity of the findings

In the conclusion section, I suggest adding a sentence about the sedentary habits reduction, also by specific advice/suggestions provided by the Government, but the limitation of virus transmission remains the first imperative condition.

Reviewer 3 ·

Basic reporting

Nothing to report.

Experimental design

Nothing to report.

Validity of the findings

Nothing to report.

Additional comments

Thank you for addressing my comments. Congratulation the article has improved, and in my mind, ready for publication.